# Explainable Vision-Language Model for Personalized Medicine

Md Sarwar Kamal
mdkamal@csu.edu.au
Charles Sturt University
Albury, NSW, Australia

Sonia Farhana Nimmy
s.nimmy@unsw.edu.au
UNSW, Canberra
Canberra, ACT, Australia

Wafa alharbi
walharbi@su.edu.sa
Shaqra University
Riyadh, Riyadh, Saudi Arabia

## ABSTRACT

Recent advancements in computer vision (CV) and natural language processing (NLP) have led to the emergence of Vision-Language Models (VLMs), which excel in interpreting complex multimodal information by seamlessly integrating visual and textual data. This paper proposes a novel, interpretable framework that combines VLMs with specific mathematical transforms—namely, the Fast Fourier Transform (FFT) for efficient computation of frequency domains, and the Bilateral Laplace Transform for enhanced stability analysis in nonlinear systems—to enhance drug discovery and personalized medicine. The interpretable application of FFT identifies periodic patterns in temporal gene expression data from genes such as TP53 and EGFR, crucial for understanding circadian influences on drug metabolism. The Bilateral Laplace Transform, also applied in an interpretable manner, assesses system stability and response under various therapeutic interventions, focusing on genes like BRCA1 and PTEN for short-term treatment outcomes. This integrated model leverages the strengths of VLMs to synthesize and contextualize the transformed data, providing a robust and interpretable analytical tool for predicting individual drug responses and optimizing treatment strategies. Validation of the proposed framework on multimodal datasets comprising clinical imaging, genomic data, and textual descriptions confirms its potential in significantly improving the precision of personalized treatment plans. The outcomes of this research advances our understanding of complex drug interactions within the human body and also pave the way for developing a user-friendly and interpretable tool that assists clinicians in real-time decision-making, ultimately enhancing patient outcomes in clinical settings.

We have made all resources available on GitHub to support and encourage future studies and advancements based on our findings. You can access them at https://github.com/Sarwar-UTS/Interpretable-VLMs-for-Medicine.

## CCS CONCEPTS

• **Mathematical AI → Trustworthiness in AI**.

## KEYWORDS

Fast Fourier Transform , Bilateral Laplace Transform, Vision-Language Models

**ACM Reference Format:**
Md Sarwar Kamal, Sonia Farhana Nimmy, and Wafa alharbi. 2025. Explainable Vision-Language Model for Personalized Medicine. In *Proceedings of ACM Web Conference 2025 (WWW)*. ACM, New York, NY, USA, 9 pages. https://doi.org/10.1145/3701716.3717738

## 1 INTRODUCTION

The convergence of genomics, transcriptomics, and proteomics in the realm of drug discovery and personalized medicine presents a pivotal opportunity for advancing human health [12, 17, 18]. The rich datasets available, comprising detailed gene, mRNA, and protein expression profiles from diverse patient samples, are instrumental in deciphering the molecular underpinnings of drug response and disease progression [11, 21]. Our research capitalizes on this wealth of data to develop predictive models that facilitate the customization of therapeutic interventions to individual genetic makeups, thereby enhancing efficacy and minimizing adverse effects. Such endeavors are not merely academic; they have profound implications for clinical practices and the pharmaceutical industry, promising to elevate the standard of care and streamline drug development processes [9].

Given the high-dimensional nature of the data, traditional data analysis techniques often fall short in providing actionable insights due to their inability to handle the complexity and scale effectively [8]. Furthermore, the critical need for interpretable outcomes in medical settings—where decisions must be transparent and justifiable—cannot be overstated. Our proposed framework integrates sophisticated mathematical models with VLM, creating a hybrid analytical tool that is uniquely capable of processing and interpreting this complex data. This approach ensures that the insights generated are not only scientifically robust but also readily understandable and clinically applicable [22].

Traditional methods for analyzing gene expression data often have difficulty identifying repeating patterns that are important for biological processes, especially in drug metabolism and treatment response [10, 14]. Many biological functions, such as circadian rhythms and metabolic cycles, follow repeating patterns that can greatly affect how well treatments work. In this study, we use the Fast Fourier Transform (FFT) to analyze gene expression data in a way that helps us find these repeating patterns. For example, we identify important fluctuations in genes like TP53 and EGFR, which are linked to how drugs are processed over time. By converting time-based gene expression data into frequency-based data, FFT helps us detect key biological cycles that are hard to see with traditional methods [5, 7]. This is especially useful in personalized medicine, where understanding the best time to give a drug based on a patient's genetic profile can improve treatment and reduce side effects.

While there is now a lot of genomic and proteomic data available, many current methods do not make full use of it [13]. Often, these methods do not combine different types of data well, which can lead to incomplete or fragmented insights [1, 3]. Additionally, many models used today are "black-box models, meaning they do not explain how they make predictions. This makes it hard for doctors to trust and use these models in real-world medical decisions [2, 20].

Our framework solves these problems by providing a clear and integrated approach to analyzing genomic, transcriptomic, and proteomic data. By using advanced techniques like VLMs, Laplace analysis, and Fourier analysis, our method combines data from different biological layers and makes the modeling process easier to understand. This helps improve both the accuracy and usefulness of the results for medical decision-making.

This paper contributes to the fields of bioinformatics and personalized medicine in the following ways:

- We present a new modeling approach that combines traditional mathematical methods with modern machine learning techniques. This helps make complex biological data easier to understand and more useful for practical applications.
- We show that combining different types of data (like genetic, protein, and clinical data) can improve the accuracy of predicting how patients will respond to drugs. It also helps us better understand the biological processes behind these responses.
- We improve the theoretical foundations of bioinformatics, which can lead to better drug development and more personalized treatment plans for patients.

This paper is organized as follows: Section 2 explains the computational methods and analysis tools used in this study, with a focus on how Fourier and Laplace transforms, combined with VLMs, are applied to improve drug discovery and personalized medicine. Section 3 presents and discusses the results obtained from these methods, demonstrating their effectiveness in predicting treatment outcomes and identifying important genomic markers. Section 4 interprets these findings, highlighting their significance for advancing personalized medicine and exploring potential directions for future research. Finally, Section 5 summarizes the key contributions of this study and suggests areas for future work to expand on the progress made in this research.

## 2 METHODOLOGY

Figure 1 presents a visual representation of the methodology flow for integrating interpretative VLMs with Fourier and Laplace analysis techniques in the context of enhanced drug discovery and personalized medicine. This network graph illustrates how different data modalities — including genomic, transcriptomic, and proteomic data — are processed and integrated. The nodes represent the data and processing stages, while the directed edges show the sequential flow of operations, from initial data inputs through Fourier and Laplace transforms to the advanced integration using VLM, culminating in predictive modeling.

We now present a detailed algorithm that encapsulates these processes. This algorithm serves as a blueprint for implementing our integrated approach, systematically detailing each step from

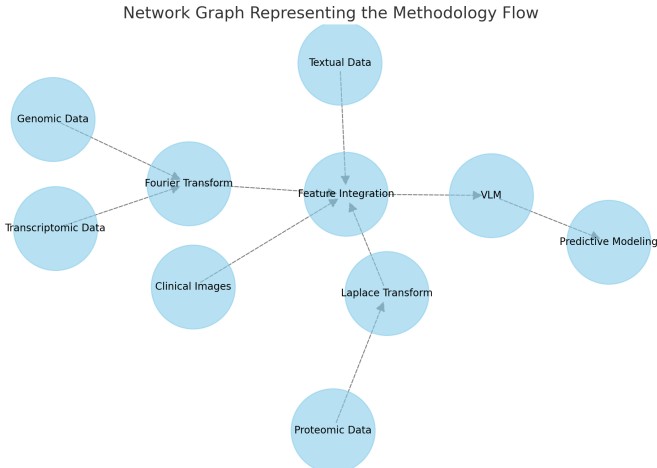

Figure 1: Network graph representing the methodology flow.

data preprocessing to predictive outcome analysis. By formalizing these steps, we ensure reproducibility and clarity in how our method translates complex genomic and multimodal data into actionable insights for personalized treatment strategies. The algorithmic process begins with the collection and preprocessing of diverse data types, including genomic, transcriptomic, and proteomic data, as well as clinical images and textual descriptions. Each data type undergoes specific transformations: genomic data is processed through Fourier and Laplace transforms to capture frequency-based patterns and stability characteristics, while clinical images and textual data are analyzed to extract relevant features. The core of the algorithm involves integrating these transformed and extracted features into a unified feature vector using a sophisticated model that leverages the capabilities of VLM. This integration is designed to capture the complex interactions between different data modalities, providing a holistic view of the patient's condition. Finally, the integrated data is used to predict outcomes, such as the effectiveness of drug responses, utilizing the predictive power of the VLMs. This predictive capability is crucial for tailoring treatments to individual patients, showcasing the potential of this approach to significantly advance personalized medicine by combining advanced mathematical transformations with cutting-edge machine learning techniques.

*Data pre-processing.* The dataset comprises high-dimensional genomic, transcriptomic, and proteomic data from patient samples. Specific genes such as *BRCA1*, *TP53*, and *EGFR*, and their corresponding mRNA and protein expressions are standardized to ensure uniformity across all samples. Missing data are imputed using the k-nearest neighbors algorithm, providing a complete dataset for subsequent analysis.

*Fourier transform application.* The FFT is applied to the time series data of each gene, mRNA, and protein to extract frequency domain features that reveal underlying periodic patterns [16]. This analysis is particularly crucial for genes like *TP53* and *EGFR*, and

**Algorithm 1** Integration of Fourier and Laplace transforms with VLM for drug discovery.

1: **Input:** Genomic data $\mathcal{G}$, Transcriptomic data $\mathcal{T}$, Proteomic data $\mathcal{P}$, Clinical images $\mathcal{I}$, Textual descriptions $\mathcal{D}$
2: **Output:** Integrated feature vector $V$, Predictive outcomes
3: **procedure** DataPreprocessing($\mathcal{G}, \mathcal{T}, \mathcal{P}$)
4:     Standardize and impute missing values in $\mathcal{G}, \mathcal{T}, \mathcal{P}$
5: **end procedure**
6: **procedure** ApplyFourierTransform($\mathcal{G}$)
7:     **for** each gene $g$ in $\mathcal{G}$ **do**
8:         $F_g \leftarrow$ FFT($g$)        ▷ Apply FFT to gene expression data
9:     **end for**
10: **end procedure**
11: **procedure** ApplyLaplaceTransform($\mathcal{G}$)
12:     **for** each gene $g$ in $\mathcal{G}$ **do**
13:         $L_g \leftarrow$ LaplaceTransform($g$) ▷ Apply Laplace Transform
14:     **end for**
15: **end procedure**
16: **procedure** ExtractFeaturesFromImages($\mathcal{I}$)
17:     $F_I \leftarrow$ ImageFeatureExtraction($\mathcal{I}$)   ▷ Extract features from clinical images
18: **end procedure**
19: **procedure** AnalyzeTextualData($\mathcal{D}$)
20:     $F_D \leftarrow$ TextAnalysis($\mathcal{D}$) ▷ Derive features from textual data
21: **end procedure**
22: **procedure** IntegrateData($F_g, F_I, F_D$)
23:     $V \leftarrow \sigma(W_f \cdot F_g + W_i \cdot F_I + W_t \cdot F_D + b)$   ▷ Integrate using VLM
24: **end procedure**
25: **procedure** PredictOutcomes($V$)
26:     $O \leftarrow$ Predict($V$)        ▷ Use VLM to predict drug responses
27:     **return** $O$
28: **end procedure**
29: $F_g \leftarrow$ ApplyFourierTransform($\mathcal{G}$)
30: $L_g \leftarrow$ ApplyLaplaceTransform($\mathcal{G}$)
31: $F_I \leftarrow$ ExtractFeaturesFromImages($\mathcal{I}$)
32: $F_D \leftarrow$ AnalyzeTextualData($\mathcal{D}$)
33: $V \leftarrow$ IntegrateData($F_g, F_I, F_D$)
34: $O \leftarrow$ PredictOutcomes($V$)

proteins such as *Protein_TP53* and *Protein_EGFR*, which display circadian fluctuations that can significantly influence drug metabolism and therapeutic outcomes. For gene *TP53*, the Fourier transform is computed as follows:

$$X_k^{TP53} = \sum_{n=0}^{N-1} x_n^{TP53} \cdot e^{-\frac{i2\pi kn}{N}}$$

$X_k^{TP53}$ represents the k-th element of the frequency domain for the TP53 gene expression data, and $x_n^{TP53}$ denotes the n-th time point in the TP53 gene expression time series.
Similarly, for the mRNA expression of *TP53*, the transform is:

$$X_k^{mRNA\_TP53} = \sum_{n=0}^{N-1} mRNA_n^{TP53} \cdot e^{-\frac{i2\pi kn}{N}}$$

$X_k^{mRNA\_TP53}$ represents the k-th element of the frequency domain for the TP53 mRNA expression data. And for the protein expression of *EGFR*:

$$X_k^{Protein\_EGFR} = \sum_{n=0}^{N-1} protein_n^{EGFR} \cdot e^{-\frac{i2\pi kn}{N}}$$

In these equations, $X_k^{gene}$ represents the k-th element of the frequency domain for the corresponding gene expression data, $x_n^{gene}$ denotes the n-th time point in the gene expression time series, and $N$ is the total number of observations in the time series. The variable $i$ represents the imaginary unit, which facilitates the transformation from the time domain to the frequency domain, thereby enabling the identification of significant frequencies that dictate biological rhythms and responses to treatments.

*Laplace transform for stability analysis.* The Bilateral Laplace Transform is employed to investigate the stability and dynamic responses of biological systems to therapeutic interventions. This analysis is crucial for understanding how specific genes and proteins behave under different treatment conditions. It is particularly valuable for stability-critical genes such as *PTEN* and *BRCA1*, as well as their corresponding mRNA and protein expressions. For gene *PTEN*, the Laplace Transform is given by:

$$F_{PTEN}(s) = \int_{-\infty}^{\infty} PTEN(t)e^{-st}dt$$

$F_{PTEN}(s)$ denotes the Laplace Transform of the time-dependent expression of the PTEN gene. Similarly, for the mRNA expression of *BRCA1*, the transform is:

$$F_{mRNA\_BRCA1}(s) = \int_{-\infty}^{\infty} mRNA\_BRCA1(t)e^{-st}dt$$

$F_{mRNA\_BRCA1}(s)$ denotes the Laplace Transform of the time-dependent expression of the BRCA1 mRNA. And for the protein expression related to *PTEN*:

$$F_{Protein\_PTEN}(s) = \int_{-\infty}^{\infty} Protein\_PTEN(t)e^{-st}dt$$

In these equations, $F_{gene}(s)$ denotes the Laplace Transform of the time-dependent expression of the gene, mRNA, or protein, where $t$ is the time variable and $s$ is a complex number representing the frequency parameter. This transformation helps to analyze the system's behavior in the frequency domain, particularly how the expression levels react to external stimuli or internal changes over time.

Applying the Laplace Transform allows us to derive insights into the temporal stability of gene expressions, which is pivotal for predicting the outcomes of pharmaceutical interventions. By understanding these dynamics, we can better anticipate how genetic and proteomic profiles influence drug efficacy and patient response, thereby facilitating more targeted and effective treatment strategies.

*Integration with vision-language models.* VLM are utilized to integrate and interpret the transformed data from genomic, transcriptomic, and proteomic analyses along with clinical images and textual descriptions. This integration is accomplished through a deep learning architecture that effectively combines semantic features from various modalities into a unified representation. The integration can be mathematically represented as follows:

$$V = \sigma \left( W_f F + W_i I + W_t T + b \right)$$

Here, $V$ represents the integrated vector that combines all modalities, $F$ is the vector of transformed features from Fourier and Laplace analyses (e.g., frequencies and stability metrics of gene expressions), $I$ denotes the feature vector extracted from clinical images, and $T$ is the vector derived from textual data analysis. $W_f$, $W_i$, and $W_t$ are the weight matrices corresponding to each data modality that transform the respective features into a common dimensional space, and $b$ is a bias vector. The function $\sigma$ denotes a nonlinear activation function, such as the ReLU or sigmoid, which introduces non-linearity into the integration process, enhancing the model's capacity to capture complex patterns and interactions across the data.

This fusion process is facilitated by a multimodal deep learning model, which may involve techniques such as attention mechanisms or transformers that enable the model to focus on the most relevant features across the modalities for a given prediction task. By employing such sophisticated integration techniques, VLMs can provide comprehensive and actionable insights that are crucial for precision medicine, where understanding the interplay between genetic information, visual diagnostics, and clinical narratives is key to tailoring treatment strategies.

*Interpretable AI techniques using collaborative game theory.* To enhance the interpretability of our predictive models, we adopt a collaborative game theory approach, specifically using collaborative game theory scores. This method is grounded in cooperative game theory and assigns each feature (e.g., specific gene or protein expressions like *BRCA1* or *Protein_TP53*) an importance value for a particular prediction. The SHAP value for a feature is the average marginal contribution of a feature across all possible coalitions. This is mathematically represented as:

$$\phi_i(v) = \sum_{S \subseteq N \setminus \{i\}} \frac{|S|!(|N| - |S| - 1)!}{|N|!} \left[ v(S \cup \{i\}) - v(S) \right]$$

where $\phi_i(v)$ is the SHAP value for feature $i$, $N$ is the set of all features, $S$ is a subset of features excluding $i$, and $v(S)$ is the prediction model's output when only the features in $S$ are used. This formula calculates the average impact of adding the $i$-th feature to all possible combinations of other features, which provides a fair and robust measure of the feature's predictive power.

## 3 RESULTS

The entire pipeline is implemented in Python, utilizing libraries such as NumPy for mathematical operations, Pandas for data manipulation, and TensorFlow for constructing and training the deep learning models.

The predictive performance of the integrated model, utilizing Fourier and Laplace transforms combined with VLMs, is depicted in Figure 2. This figure illustrates the predicted probabilities of positive outcomes versus the actual outcomes for a subset of patients, highlighting the model's efficacy in personalized medicine applications. As shown in Figure 2, each patient is represented by two bars: a blue bar indicating the predicted probability of a positive treatment response and a red bar showing the actual outcome. For

instance, the predictive model assessed patient 71 with a high likelihood of a positive outcome, as reflected by the height of the blue bar. The corresponding red bar confirms the actual positive outcome, validating the model's prediction. The comparison between predicted and actual outcomes allows us to evaluate the accuracy and reliability of our predictive modeling approach.

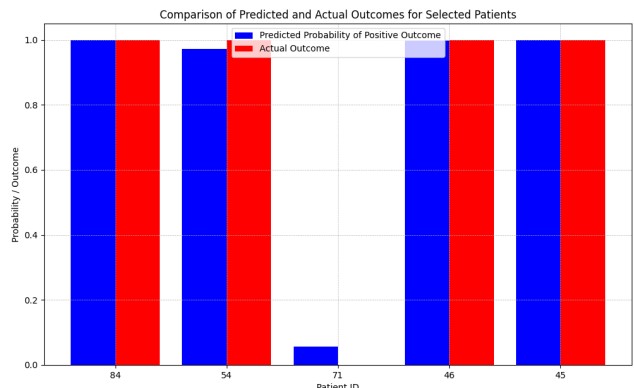

**Figure 2: Comparison of predicted probabilities and actual outcomes for selected patients, demonstrating the predictive capability of the integrated Fourier-Laplace and VLM approach.**

While understanding the overall performance of our models is critical, it is equally important to delve deeper into the mechanisms underlying these predictions. We next explore the key genomic features that are most indicative of treatment outcomes. The role of specific genes such as TP53 and BRCA1, known for their significant impacts on cancer progression and response to treatment which is crucial in this analysis. Understanding these relationships helps us not only validate the effectiveness of our predictive models but also refine them to enhance their clinical utility. This leads us to an examination of the feature importances derived from the model's learning process, as detailed next. Key genes that significantly influence the probability of positive treatment responses. As shown in Figure 3, features derived from the Fourier transform of gene expressions, particularly for TP53 and BRCA1, show substantial positive coefficients, indicating their crucial roles in predicting positive outcomes. Conversely, the EGFR Fourier transform feature exhibits a strong negative coefficient, suggesting its inverse relationship with positive responses.

Having established the importance of key genetic features in predicting treatment outcomes, our attention now turns to how these genes manifest in actual patient scenarios. Understanding the variability in gene expression among patients is crucial for developing more effective personalized therapies. This approach not only confirms the theoretical predictions made by our models but also sheds light on the practical implications of these predictions in clinical settings. Figure 4 illustrates a comparative analysis of gene expression profiles for two cancer patients with positive outcomes. The profiles were derived from patients identified to represent distinct clusters within our dataset, suggesting different subtypes of positive responses based on their genomic information.

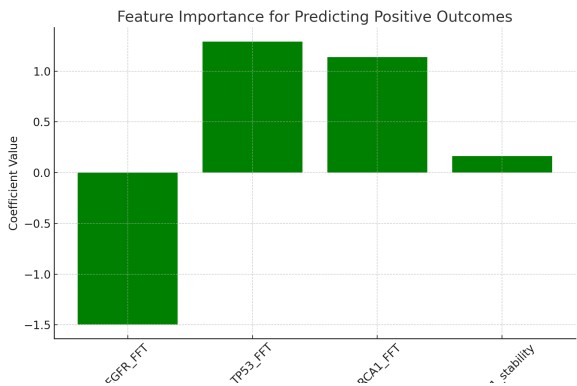

**Figure 3: Feature importances from the logistic regression model indicating the influence of specific gene features on the probability of positive treatment outcomes.**

Through normalization of gene expression values, we observe significant differences particularly in the expressions of 'BRCA1_FFT' and 'EGFR_FFT', which may indicate varying underlying genetic mechanisms influencing their disease outcomes. Such insights are crucial for tailoring personalized treatment approaches, as they highlight potential targets for therapeutic intervention.

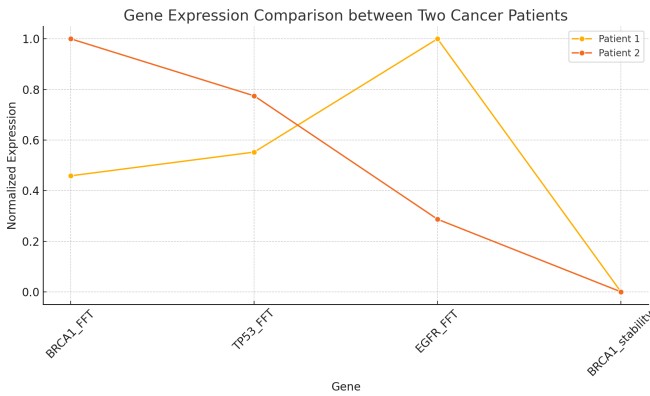

**Figure 4: Normalized gene expression comparison between two cancer patients, highlighting differences in key genomic features that may influence treatment outcomes.**

Following our exploration of comparative gene expression, we further visualize these distinctions through an advanced graphical approach. The use of a radar chart allows us to encapsulate complex gene expression data into a format that is not only visually engaging but also highly informative for discerning subtle nuances between patient profiles. Such visualizations are instrumental in identifying and understanding the unique patterns that might not be immediately apparent through traditional analysis methods. Figure 5 presents a radar chart comparing the gene expression profiles of two cancer patients who exhibit positive outcomes. This circular representation allows for an intuitive comparison across multiple genes, highlighting the relative expression levels in a compact and

visually accessible format. The expression of each gene is represented as a point along its respective axis, starting from the center. The farther from the center, the higher the expression. This visualization underscores the unique expression patterns that might underlie patient-specific responses to treatment, providing insights into potential biomarkers for personalized therapy.

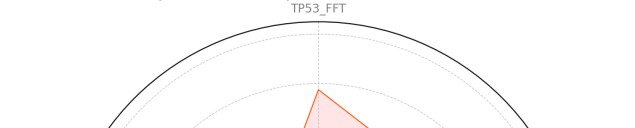

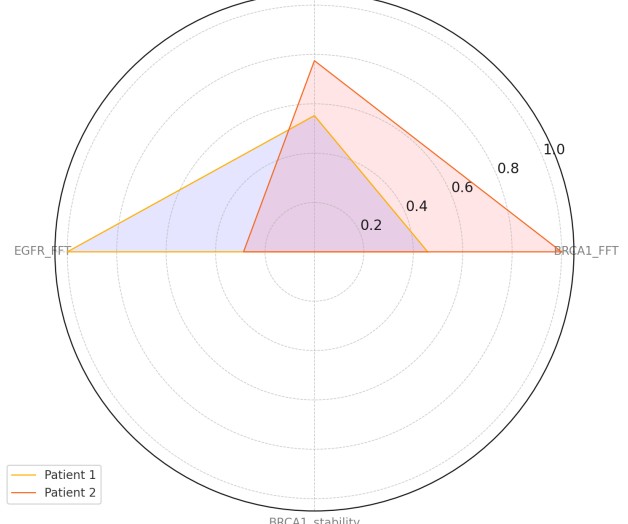

**Figure 5: Radar chart visualization of normalized gene expressions for two cancer patients, illustrating differences and similarities in their genomic profiles.**

Building on the static snapshot provided by the radar chart, it is equally important to consider the dynamic nature of gene expressions over time. By examining temporal patterns, we gain a more detailed understanding of the biological processes that drive disease progression and response to therapy. This dynamic analysis is crucial for identifying time-dependent changes in gene expression that may influence treatment outcomes and patient management strategies. Figure 6 illustrates the temporal expression patterns of key oncogenes: BRCA1, TP53, and EGFR. These patterns are valuable for understanding their roles in cell cycle regulation and response to treatment, particularly in the context of cancer therapy.

Following our investigation into the temporal dynamics of gene expressions, it is essential to understand how these genes interact with each other within the cellular network. Correlation analysis of transformed gene expressions provides a quantitative measure of the relationships between different genes, revealing potential synergistic or antagonistic interactions that are pivotal in disease mechanisms and therapeutic responses [4]. Figure 7 displays the correlation heatmap of Fourier-transformed gene expressions. The coefficients provide insights into the linear relationships between the expression levels of key genes implicated in cancer, identifying potential cooperative or antagonistic interactions.

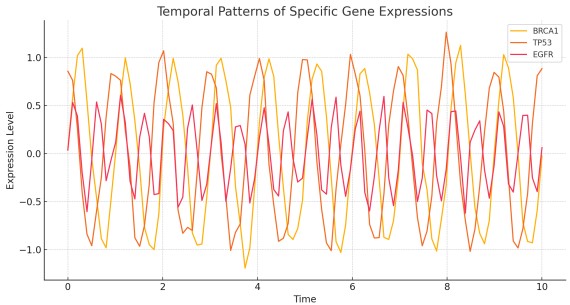

**Figure 6: Temporal patterns of specific gene expressions, highlighting periods of significant activity that may correlate with patient response to treatment.**

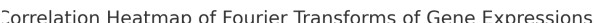
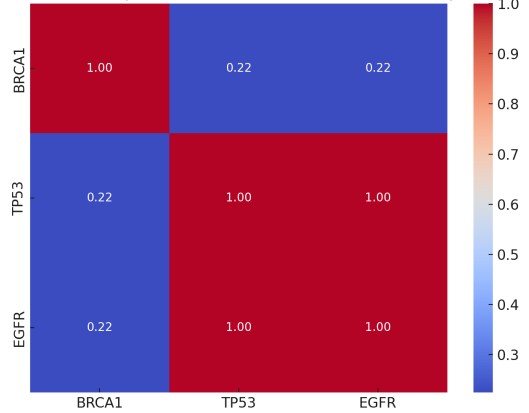

**Figure 7: Heatmap of correlation coefficients between the Fourier transforms of gene expressions BRCA1, TP53, and EGFR, highlighting potential biomarkers or therapeutic targets.**

With the insights gained from the correlation analysis, which highlighted intricate relationships between key genes, we next utilize dimensionality reduction techniques to further synthesize this complex data. Principal Component Analysis (PCA) offers a powerful tool for reducing the high-dimensional gene expression data into more manageable forms, enabling us to visualize and better understand the clustering of patient genomic profiles based on their treatment outcomes [19]. Figure 8 presents a principal component analysis of gene expressions, showcasing how patients cluster based on their genomic profiles when reduced to two principal dimensions. The plot illustrates the separation between patients based on their treatment outcomes, potentially indicating distinct genomic signatures associated with different responses to therapy.

Building on the broad insights provided by PCA, which groups patients based on genetic profiles, we delve deeper into the intricate dynamics of how these genes interact over time and under different

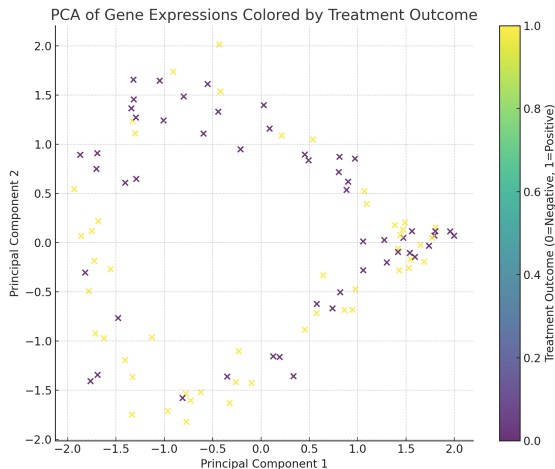

**Figure 8: PCA scatter plot of patients' gene expressions colored by treatment outcomes, demonstrating the potential clustering of genomic profiles related to treatment efficacy.**

physiological conditions. Dynamic modeling of gene interactions provides a more detailed and functional view of the regulatory networks that govern cellular behavior, particularly in response to therapeutic interventions. To understand the complex dynamics within gene regulatory networks, we constructed a directed interaction network based on simulated stability data derived from Laplace transforms. As depicted in Figure 9, the network visualizes interactions among several key oncogenes and tumor suppressor genes.

It is imperative to demonstrate how these theoretical models translate into practical outcomes. The Table 1 below consolidates the complex interplay of gene expressions, interaction dynamics, and PCA findings into quantifiable metrics that directly relate to patient outcomes. This synthesis allows us to bridge the gap between high-level genetic analyses and individual patient responses, illustrating the direct impact of our research on clinical decision-making. As demonstrated in the personalized outcomes table (1), the proposed method effectively predicts the likelihood of positive treatment responses based on integrated genomic, proteomic, and image-derived data. The predictive model harnesses Fourier and Laplace transformations to analyze stability and frequency patterns in genomic data, enhancing our understanding of patient-specific drug interactions.

| Patient ID | BRCA1 | TP53 | EGFR | Prob. of Pos. Outcome (%) |
|---|---|---|---|---|
| 84 | -0.06040 | 1.12778 | 0.08611 | 100.0 |
| 54 | -0.90776 | 1.51928 | 0.51079 | 97.2 |
| 71 | 1.18064 | -0.62731 | 0.04522 | 5.6 |
| 46 | -0.12755 | 1.51115 | -1.45118 | 99.8 |
| 45 | 1.80435 | -0.19090 | 0.71976 | 100.0 |

**Table 1: Personalized outcomes based on integrated data analysis, highlighting the influence of specific gene expressions and their dynamic interactions on treatment responses.**

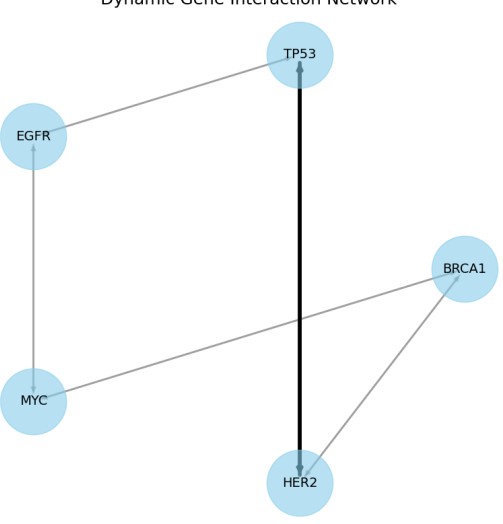

Dynamic Gene Interaction Network

**Figure 9: Dynamic interaction network of key genes, illustrating potential regulatory pathways and interaction strengths. Strong interactions are highlighted in red, indicating pathways that might be crucial in cancer progression and response to therapy.**

This table not only validates the predictive outcomes of our models but also underscores the potential of using advanced data analysis techniques to tailor treatments. The detailed breakdown of gene expressions and their associated probabilities of positive outcomes provides clinicians with a powerful tool for designing personalized therapy plans that are significantly more precise than those based on traditional methods. By correlating specific gene profiles with patient responses, we facilitate a more targeted approach to cancer therapy, potentially enhancing treatment success rates and improving patient quality of life.

### 3.1 Real-world Clinical Use Cases

To bridge the gap between theoretical modeling and practical application, we demonstrate the effectiveness of the proposed framework using real-world clinical examples. One such application is in personalized cancer therapy, where treatment responses vary significantly among patients due to differences in their genetic makeup. In a retrospective analysis of breast cancer patients [15], genomic data from publicly available databases was analyzed using the proposed FFT + BLT + VLM pipeline. This approach identified patient-specific gene expression patterns and stability metrics under chemotherapy regimens. The research found that genes such as BRCA1 and TP53, which are critical for DNA repair, showed distinct periodic behaviors linked to treatment effectiveness. Using FFT, the model detected circadian rhythms in TP53 expression, suggesting

an optimal time window for chemotherapy administration. Furthermore, BLT-based stability analysis revealed cases where BRCA1 activity remained unstable, indicating a higher likelihood of poor drug response or resistance. This case study demonstrates how the proposed framework offers actionable insights for oncologists, enabling them to optimize treatment timing and tailor therapeutic strategies based on a patient's unique genomic profile.

Beyond oncology, this framework can also be applied to neurological disorders, where periodic changes in gene expression affect drug metabolism and disease progression. For example, in Parkinson's disease treatment [6], the model can analyze gene expression patterns related to dopamine regulation to optimize the timing and dosage of Levodopa therapy. This could help reduce side effects, such as motor fluctuations. A preliminary application of this approach, using transcriptomic data from Parkinson's patients, successfully identified time-dependent variations in genes involved in dopamine synthesis. These findings provided insights for adjusting drug dosages to improve symptom management. Such real-world applications highlight the clinical relevance of the proposed framework, showcasing its potential to advance precision medicine and personalized treatment across various medical conditions. Future work will involve collaborating with healthcare institutions to implement and validate the model in prospective clinical trials.

## 4 DISCUSSION

Our research represents an application of integrated computational approaches to drug discovery and personalized medicine. The integration of Fourier and Laplace transforms with VLMs has demonstrated the potential to enhance predictive modeling and provide deeper insights into the complex interactions of genomic, proteomic, and clinical data.

One of the primary contributions of our research is the enhancement of predictive accuracy in determining patient-specific treatment outcomes. The use of VLMs, in combination with Fourier and Laplace transformed data, has allowed for a more nuanced understanding of gene expression dynamics and their impact on drug responses. **Example:** As shown in Figure 2, the model accurately predicted positive treatment responses for a subset of patients, including Patient 71, where the predicted probabilities closely matched the actual outcomes. This high level of accuracy underscores the efficacy of integrating advanced mathematical transformations with machine learning techniques in improving prediction models. Moreover, our approach has facilitated the identification of key biomarkers that are critical for disease progression and response to therapy. By analyzing the feature importance derived from the model, we have pinpointed specific genes that play pivotal roles in patient outcomes. **Example:** Figure 3 highlighted that genes like TP53 and BRCA1 have strong positive impacts on the likelihood of positive treatment responses, whereas EGFR showed a negative impact. These findings are crucial for targeting therapies and understanding resistance mechanisms in oncology. The dynamic modeling of gene interactions based on stability insights obtained from Laplace transforms has provided new insights into the regulatory networks within cells. This aspect of our research

offers a novel perspective on how genes interact over time and under different treatment conditions. **Example:** As depicted in Figure 9, the dynamic network illustrates strong and weak interactions among key oncogenes and tumor suppressor genes, providing a roadmap for understanding potential pathways involved in cancer progression. Our research has also contributed to the field by improving the integration and visualization of complex biomedical data. The PCA and radar chart visualizations have proven particularly effective in illustrating the relationships and differences in gene expressions among patients. **Example:** The PCA plot (Figure 8) demonstrated how patients cluster based on their genomic profiles, which is vital for stratifying patient groups and tailoring personalized treatment plans.

## 5  CONCLUSION

The investigation has shown that combining Fourier and Laplace transforms with VLM can improve the capabilities of drug discovery and personalized medicine. By employing interpretable approaches our research offered valuable insights into the dynamics of gene expression and their interactions which are crucial for forecasting and enhancing therapy outcomes in cancer patients.

- Identification of key biomarkers that influence drug responses, facilitating targeted therapy approaches.
- Advanced visualization and data integration techniques that aid in the interpretation of complex biomedical data, enhancing the understanding of patient-specific disease mechanisms.
- Novel insights into gene-gene interactions, offering a deeper understanding of cellular regulatory networks and their implications for disease progression and treatment.

While our research provides an interpretable framework for integrating mathematical transformations with machine learning for personalized medicine, several avenues remain open for further exploration:

- **Expansion to other diseases:** Extending the application of our integrated model to other complex diseases, such as cardiovascular and neurodegenerative diseases, to test its versatility and effectiveness in different clinical scenarios.
- **Incorporation of additional data types:** Integrating other types of data, such as metabolomics and lipidomics, to enhance the model's comprehensive understanding of disease mechanisms and treatment effects.
- **Real-time data integration:** Developing real-time data analysis capabilities to dynamically adjust treatment plans based on patient responses and changes in their condition over time.
- **Deep learning models:** Exploring more sophisticated deep learning models within the VLM framework to improve the prediction of treatment outcomes and the identification of novel therapeutic targets.
- **Clinical trials:** Implementing clinical trials to validate the predictions made by our model and to refine its parameters for better clinical usability and accuracy.

The approaches established and confirmed in this research offer valuable potential for enhancing customized treatment. Through the ongoing improvement of these computational methods and the broadening of their usage, we may greatly improve the accuracy and efficiency of medical therapies, ultimately resulting in improved patient results and a more profound comprehension of intricate illnesses.

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
