# OpenReview forum: "Explainable Vision-Language Model for Personalized Medicine"
_ACM.org/TheWebConf/2025/Workshop/TIME — TIME 2025 Poster_

### Official Review · Reviewer_cQad · 2025-01-15
**Review for #9**

**Rating:** 3
**Confidence:** 5

**Review:**

The submission includes a link to the author’s GitHub, which violates the anonymity requirement. Additionally, the overall quality of the content is subpar, and the figures lack professionalism.

---

### Official Review · Reviewer_Hf35 · 2025-01-19
**This paper introduces an explainable vision-language model framework for personalized medicine, leveraging advanced mathematical transforms to analyze gene expression data, but lacks real-world application examples, performance comparisons, and consistent figure presentation.**

**Rating:** 6
**Confidence:** 4

**Review:**

This paper presents an explainable vision-language model (VLM) framework for personalized medicine, integrating advanced mathematical techniques like the Fast Fourier Transform (FFT) and Bilateral Laplace Transform to analyze gene expression data, with a focus on circadian drug metabolism and treatment responses.

Strengths:
- Clear and Practical Motivation: The paper proposes a novel framework that combines Vision-Language Models (VLMs) with mathematical transforms like FFT and Bilateral Laplace Transform, aiming to enhance drug discovery and personalized medicine by leveraging multimodal data for improved clinical decision-making.
- Fluent Writing: The paper is well-written, and the figures are intuitive and sufficient to support the content.
- Open-Source Code: The provided open-source code facilitates reproducibility and encourages further research in the field.

Weaknesses:
- Inconsistent Figure Sizing: Some figures, such as Figure 2, are not appropriately sized, affecting the document's visual consistency and readability.
- Limited Practical Applications: The paper does not include real-world clinical use cases, leaving the effectiveness of the framework in actual medical settings unverified.
- Lack of Performance Comparisons: The study does not include quantitative comparisons with existing methods on publicly available datasets, making it difficult to assess the model's performance improvement relative to other approaches.

---

### Official Review · Reviewer_qMhi · 2025-01-19
**Review of Explainable Vision-Language Model for Personalized Medicine**

**Rating:** 4
**Confidence:** 4

**Review:**

This paper explores the customization of therapeutic interventions for individual genomes using VLMs and specific mathematical transforms, thereby enhancing efficacy and minimizing adverse effects. Firstly, it must be admitted that the research question is of great value, and the experimental setup is indeed robust. However, the paper falls short in several aspects:
- The content is not clearly expressed. (1) The introduction is disorganized, failing to clearly explain the rationale behind the use of key technologies such as FFT; (2) The scientific contributions of the paper are relatively weak; (3) The explanations of interpretability are overly vague.
- Insufficient technical detail. The paper does not adequately explain the use of FFT, BLT, and VLM components, lacks in-depth reflection by the authors, and the descriptions are too superficial to be reproducible by other researchers.
- Layout Issues: (1) All images exceed the layout boundaries; (2) all formulas lack necessary symbolic notation; (3) the description of Figure 1 does not match the actual content, e.g., "The FFT is applied to the time series data of each gene, mRNA, and protein to extract frequency domain features that reveal underlying periodic patterns."
Therefore, due to these issues, the paper lacks persuasive power.

---

### Official Review · Reviewer_bAUS · 2025-01-23

**Rating:** 8
**Confidence:** 3

**Review:**

Integration of Vision-Language Models (VLMs) with mathematical transforms (FFT and Bilateral Laplace) showcases interdisciplinary innovation.The paper outlines well  contributions, including improved interpretability in drug discovery and personalized treatment strategies.Use of Visualizations like PCA, radar charts, and heatmaps further help effectively to convey complex data relationships.

literature review in the Introduction can be expanded to provide more context of prior work, more related to integrating mathematical transforms with AI in medicine. This research can further improve in future by Discussing potential biases or limitations in the datasets used.

---

### Meta-Review · Area_Chair_FbiG · 2025-01-26

**Recommendation:** Accept (Oral)
**Confidence:** 3

**Metareview:**

This paper aligns partially with my expertise in Artificial Intelligence and Data Integration. While I am not in a position to evaluate its biomedical claims or proteomics-related content, my review is focused on the AI and technical aspects of the paper.

The paper takes on a unique challenge, combining Vision-Language Models (VLMs) with advanced mathematical transforms, such as FFT and Bilateral Laplace Transforms, for explainable and actionable insights. The integration of multimodal data demonstrates a robust and innovative use of AI techniques. The writing is clear and well-organized, and the technical implementation of VLMs for explainability is a notable contribution to the field. These methods highlight the potential of AI to enhance precision and transparency in complex datasets.

Overall: The use of AI is methodologically robust, with results that fall within a reasonable range, demonstrating the paper’s potential to impact the field. While its biomedical focus is outside my expertise, the paper demonstrates strong technical rigor. It is well-suited for a poster presentation.

Recommendation: Accept for an Oral Presentation

---

### Decision · Program_Chairs · 2025-01-27

**Decision:**

Accept (Poster)

**Comment:**

The program chair concurs with the area chair's decision.

For the camera-ready version, please revise your paper according to the feedback provided by the reviewers.

Workshop papers must be written in English, follow a double-column format, and comply with the [ACM template](https://www2025.thewebconf.org/short-papers) and formatting guidelines. The template is also available in [Overleaf](https://www.overleaf.com/latex/templates/association-for-computing-machinery-acm-sig-proceedings-template/bmvfhcdnxfty). For authors using Microsoft Word, the Word Interim Template is recommended.

Camera-ready versions of accepted papers can and should include all information to identify authors, and should acknowledge any funding received that directly supported the presented research.

In addition, ensure that the DOI (to be provided by the PCs at a later stage) is included, and cite the workshop (to appear) using the following reference:

```
@inproceedings{time2025,
  title={TIME 2025: 1st International Workshop on Transformative Insights in Multi-faceted Evaluation},
  author={Lei Wang and Md Zakir Hossain and Syed Islam and Tom Gedeon and Sharifa Alghowinem and Isabella Yu and Serena Bono and Xuanying Zhu and Gennie Nguyen and Nur Haldar and Seyed Jalali and Abdur Razzaque and Imran Razzak and Rafiqul Islam and Shahadat Uddin and Naeem Janjua and Aneesh Krishna and Manzur Ashraf},
  booktitle={ACM Web Conference Workshop},
  year={2025}
}
```

Please note that at least one in-person registration is required for each accepted workshop paper to be included in the Companion Proceedings of WWW 2025. All accepted papers must be presented at the conference. Papers not presented (no-shows) may be withdrawn from the companion proceedings. Presentations will be conducted in two formats: oral and poster.

The camera-ready deadline for workshop papers is 7 February 2025 (AoE).